# Cell-Based Mechanosensation, Epigenetics, and Non-Coding RNAs in Progression of Cardiac Fibrosis

**DOI:** 10.3390/ijms21010028

**Published:** 2019-12-19

**Authors:** Silvia Ferrari, Maurizio Pesce

**Affiliations:** 1Unità di Ingegneria Tissutale Cardiovascolare, Centro Cardiologico Monzino, IRCCS, Milan 20138, Italy; silvia.ferrari@ccfm.it; 2PhD Program in Translational Medicine, Università degli studi di Pavia, Pavia 27100, Italy

**Keywords:** mechanotransduction, cardiac fibrosis, epigenetics, non-coding RNAs, cardiac fibroblast, heart failure

## Abstract

The heart is par excellence the ‘in-motion’ organ in the human body. Compelling evidence shows that, besides generating forces to ensure continuous blood supply (e.g., myocardial contractility) or withstanding passive forces generated by flow (e.g., shear stress on endocardium, myocardial wall strain, and compression strain at the level of cardiac valves), cells resident in the heart respond to mechanical cues with the activation of mechanically dependent molecular pathways. Cardiac stromal cells, most commonly named cardiac fibroblasts, are central in the pathologic evolution of the cardiovascular system. In their normal function, these cells translate mechanical cues into signals that are necessary to renew the tissues, e.g., by continuously rebuilding the extracellular matrix being subjected to mechanical stress. In the presence of tissue insults (e.g., ischemia), inflammatory cues, or modifiable/unmodifiable risk conditions, these mechanical signals may be ‘misinterpreted’ by cardiac fibroblasts, giving rise to pathology programming. In fact, these cells are subject to changing their phenotype from that of matrix renewing to that of matrix scarring cells—the so-called myo-fibroblasts—involved in cardiac fibrosis. The links between alterations in the abilities of cardiac fibroblasts to ‘sense’ mechanical cues and molecular pathology programming are still under investigation. On the other hand, various evidence suggests that cell mechanics may control stromal cells phenotype by modifying the epigenetic landscape, and this involves specific non-coding RNAs. In the present contribution, we will provide examples in support of this more integrated vision of cardiac fibrotic progression based on the decryption of mechanical cues in the context of epigenetic and non-coding RNA biology.

## 1. Introduction: Relevance of Cell Mechanics in Cardiac Fibrosis

Assessment of the mechanically dependent molecular machinery has become a new, insightful approach to deciphering cellular dynamics inside tissues, with implications in morphogenesis, tissue renewal, and pathology progression. For example, mechanically dependent coordination of tissue growth is recognized as a major determinant, either in pre-implantation or post-implantation embryonic patterning [1], as well as in cellular pathologic evolutions, such as in cancer [2]. Moreover, progenitor cell differentiation has been directly linked to compliance of the extracellular matrix (ECM) [3] and to geometry-dependent intracellular traction forces transmitted through the cytoskeleton to intracellular compartments, for example, the nucleus [4,5,6,7,8].

The heart is an organ undergoing continuous motion, with more than three billion contraction cycles during the average human life span. In a healthy myocardium, ECM turnover is under continuous renewal by the stromal component of the myocardium, composed of cardiac fibroblasts (CFs). Under conditions promoting metabolic alterations (e.g., hyperglycemia), inflammations such as chronic myocardial ischemia, or pressure overload, cardiac fibroblasts proliferate and evolve into profibrotic cells (so-called myofibroblasts (myoFbs), which contribute to consistent extracellular matrix deposition (prevalently, collagen I and III) and promote myocardial stiffening [9,10]. As a consequence, the Young’s elastic modulus increases up to ten times, ranging from 10–20 kPa in the physiological tissue to 50–200 kPa in the pathological tissue [11].

Other than being connected to pathological activation, cardiac matrix mechanics is relevant for controlling cardiomyocytes division and maturation. For example, it has been suggested that the beginning of the embryonic myocytes beating could be related to perturbations of mechanosensitive Ca^2+^ channels, determined by the progressive increase in matrix stiffness surrounding the cardiac progenitors before the onset of electromechanical coupling [12,13,14]. Additionally, the increase in mechanical load occurring at birth determines the maturation of the cardiac myocytes (CMs) contractile apparatus [15], causing further stiffening of the myocardium and mitotic block [16,17]. Furthermore, it has been shown that until two days after birth, mammalian hearts might regenerate by a mechanism involving (re)activation of the CMs division [18], and results obtained by treating neonatal mice with drugs able to ‘soften’ the myocardial matrix showed that the temporal window of heart regeneration can be experimentally prolonged [19]. In keeping with these findings, mammalian cardiomyocytes can be induced to back-differentiate and reactivate cell cycles by low-stiffness substrates [20], and cardiac regeneration in lower vertebrates (e.g., zebrafish) is accompanied by a transient softening of the extracellular matrix [21].

The responses of cells to mechanical cues have emerged as a general component of the chronic evolution of cardiovascular diseases and aging [22], a process with an important epigenetic component [23]. Taken together, these data suggest a tight relationship between the mechanosensitivity of CFs and downstream profibrotic cell signaling. This indicates a novel way to address myocardial fibrosis and to regulate the proliferation/maturation of the contractile cells, based on targeting cell mechanosensation-related effectors.

## 2. Alterations in Myocardial Compliance and Progression of Cardiac Fibrosis: Epigenetics Aspects

Epigenetics encompasses a series of mitotically and meiotically transmissible DNA/chromatin modifications independent of changes in the primary DNA sequence [24]. Epigenetics exerts its control of the genome functions through different covalent modifications of the chromatin, consisting of DNA methylation and histone modifications, as well as by interactions with non-coding RNA. These modifications affect the topology of the chromatin and, consequently, the accessibility of DNA to factors facilitating transcription initiation and elongation, DNA replication, recombination, and repair. We direct readers to specialized review articles on the types and the overall gene regulatory effects of DNA/chromatin epigenetic modifications in cardiovascular diseases [25]. It will be sufficient here to mention the following concepts.

**(i)** DNA methylation is defined by the covalent binding of a methyl group to the 5′ carbon of cytosine and is usually associated with gene expression repression, as it can decrease the accessibility of chromatin to DNA-binding proteins or transcription factors that are required for gene transcriptional activation [26].

**(ii)** Histones, the structural component of the chromatin, represent a target of several post-translational modifications, mainly occurring on amino acid residues of the N-terminal tails that protrude from the chromatin fibers [27]. Among these, acetylation promotes the relaxation of the chromatin structure associated with gene expression activation [28], while hypoacetylation and methylation promote chromatin condensation, resulting in gene expression repression [29].

As discussed in other contributions, epigenetic modifications can affect global gene expression by setting cellular ‘memories’ that reflect onto altered phenotypes and permanent diseased states. In the cardiovascular scenario, this appears particularly important, given the chronical nature of the disease and its main metabolic basis [30]. Similar to metabolism-dependent memories, mechanical cues can also set permanent variations in cell functions, with possible consequences for chronic pathological programming. For example, it was found that cells embedded in three-dimensional hydrogels with high mechanical compliance are in a permanent activation state, even after shifting them to media with lower stiffness [31]. Importantly, the cells cultured in high stiffness media exhibit high levels of mechanically activated YAP/TAZ complex activity (see below) and the expression of pro-calcific master transcription factor RUNX-1, thus suggesting that activation of profibrotic/pro-calcific pathways can be permanently established by exposing cells to contacts with an extracellular matrix with high mechanical compliance. Since subjecting cells to mechanical constraints determines reversible nuclear shuttling of mechanically dependent transcription factors [7] and changes in the epigenetic landscape [24], it is possible that metabolic and mechanical cues cooperate in establishing a chronically activated phenotype that contributes to cardiac fibrosis [32,33,34].

While biochemical signals responsible for epigenetic alterations occurring in fibroblast activation are well described, the effect of mechanotransduction on the epigenetic set-up underlying transition from quiescent to activated myoFbs, and vice versa, is still under investigation [35]. On the other hand, some examples of pathways potentially involved in the epigenetic setting of myo-fibroblasts activations have been already provided, as follows.

Peroxisome proliferator-activated receptor (PPARα-δ and PPARα-γ) is a nuclear transcription factor highly expressed in the heart, with anti-inflammatory and anti-proliferative properties [36]. Diep and colleagues, using an animal model of cardiac hypertrophy, demonstrated that the treatment with PPARα ligand activator (fenofibrate) is able to prevent the progression of cardiac fibrosis [37]. Interestingly, in vitro experiments showed that the administration of PPARγ ligand resulted in a decrease in the collagen synthesis by angiotensin II-stimulated cardiac fibroblasts [38]. Although the molecular mechanism responsible for the PPARs anti-fibrotic effect remains unknown, growing evidence supports the involvement of epigenetics. In particular, the Methyl CpG binding protein (MeCP2), which binds methylated DNA [39], targets the 5′ end of the *PPARγ* gene promoter, silencing its transcription, and thus promotes fibroblast differentiation into myofibroblast [40], with consequences such as stiffening of the myocardial matrix.

Recent studies have shown that nuclear shape and stiffness controls gene expression and drives cell differentiation by exposing chromatin to mechanical cues and possible direct/indirect reshuffling of transcriptional accessibility [41]. These forces originate from tension forces of the cytoskeleton, namely, the actomyosin stress fibers, which transfer tensional forces from focal adhesions firmly attaching cells to stiff substrates in the nucleus through the Linker of Nucleoskeleton and Cytoskeleton (LINC) complex, nuclear lamins, and, finally, chromatin. As a result, the chromatin stretches and its accessibility to RNA polymerase II changes. Growing evidence supports the hypothesis that myofibroblast pathological activation is influenced by a convergence of mechanical stimuli transduced in the nucleus by the cytoskeleton, resulting in activation of profibrotic signaling [42]. For example, Alisafaei and colleagues used a mechanically dependent model of transcription and epigenetic factors (HDAC3, MKL) nucleocytoplasmic shuttling to investigate the actomyosin-dependent translocation in the nucleus of NIH 3T3 mice fibroblast cells. Results showed that cells placed on smaller substrate areas decreased nuclear volumes and increased histone acetylation levels, and that the reduction of actomyosin contractility caused a nuclear translocation of histone deacetylase 3 (HDAC3). Once reaching the nucleus, HDAC3 exerted its function, reducing the histone acetylation level and resulting in chromatin compaction and reduction of nuclear volume. In order to investigate the association between alterations in actomyosin contractility and HDAC3 shuttling from the cytoplasm to the nucleus, where it is responsible for chromatin condensation, Jain and colleagues studied treated NIH 3T3 mice fibroblasts with inhibitors of the actomyosin fibers contractility (e.g., Blebblistatin and Y27632) [43]. The administration of these drugs favored degradation of the nuclear factor IκB-α (which normally sequesters HDAC3 in the cytoplasm), releasing the acetylating enzyme that translocates into the nucleus. Taken together, these data suggest a correlation between the actomyosin contractility and the nucleocytoplasmic translocation of the epigenetic factors on substrates with different shapes and areas. This interplay between mechanical forces and epigenetics may greatly contribute in the activation/silencing of profibrotic pathways, establishing a direct link between mechanosensation and permanent epigenetic changes (Figure 1).

The YAP/TAZ complex has been identified as the transcriptional component of the so-called hippo pathway, an intracellular signaling cascade involved in tissue growth and homeostasis [44]. It has been demonstrated that transmission of mechanical forces to the nucleus from the cytoskeleton via the LINC complex determines YAP/TAZ translocation through the nuclear pores [6]. The involvement of YAP signaling in fibrotic processes has been demonstrated with the finding that human cardiac fibroblasts exposed to cyclic straining determines YAP nuclear translocation and activation of cell cycle genes (Figure 2) [44]. In addition, in a mouse model of myocardial infarction, it was demonstrated that YAP/TAZ is abundantly expressed in cells located juxtaposed to the infarct border zone, suggesting a direct role in control of collagen deposition and thus in ECM stiffening [45]. Moreover, the stiffening of the matrix as a consequence of myofibroblast activity mechanoactivates YAP, which positively regulates the production of profibrotic mediators and ECM proteins, resulting in a feed-forward loop of fibroblast activation and tissue fibrosis [46]. Since YAP functions are controlled by post-translational modifications such as acetylation [47], and are subject to metabolic and nutrient control [48,49]; these data establish a possible convergence between metabolic control of epigenetics and CF mechanics in cardiac fibrosis.

Other pathways, such as those related to cellular redox control, may represent potent pathological activators connected to cell mechanics in cardiac fibrosis. In this respect, Hata and colleagues observed that treating HeLa cells with agents that damaged the DNA and formed methylated bases with cytotoxic properties (alkylating agents, e.g., methyl methanesulfonate, cisplatin, doxorubicin, and *N*-methyl-*N*-nitrosourea) determined YAP translocation and decreased phosphorylation levels. Once in the nucleus, nuclear acetyltransferases cardiopulmonary bypass (CBP) targeted YAP at Lys-494 and Lys-497, determining its transcriptional activation [50]. Although a similar mechanism has not yet been demonstrated in CFs, this evidence suggests synergism between the epigenetic post-translational modification pathway and the regulation of the transcriptional component of the hippo pathway, which depends on the balance between the cytoplasmic and the nuclear-localized YAP, thus reinforcing mechanical signaling by shifting the balance in favor of nuclear translocation and activation of profibrotic genes.

Inflammation is involved in cardiac fibrosis progression at multiple levels, from the response to acute/chronic damages (e.g., ischemia and pressure overload) to the chronic setting of fibrosis, mainly consisting of collagen deposition. During the first phase after myocardial damage, the release of pro-inflammatory mediators recruits inflammatory cells, such as neutrophils and monocytes, into the myocardium and activates myocardial-resident fibroblasts [51]. Although there is no evidence of cooperation between mechanosensing-activated pathways and epigenetic circuitries in the recruitment and pro-inflammatory activities of innate immunity cells, it is important to highlight that functional cooperativity between inflammatory cells (e.g., macrophages) and fibroblasts has been recently described with the demonstration that deformations in collagen matrices produced by fibroblasts are directly involved in macrophage migration [52]. This suggests that epigenetic alterations of the fibroblasts’ activity, such as those observed in cardiac fibrosis, may supervise inflammatory cells’ recruitment by long-range mechanical signaling spread over the extracellular matrix of the damaged myocardium.

During the relapse phase of the inflammatory process, a clearer relationship may exist between epigenetics CFs mechanics and fibrosis through the effects of the transforming growth factor-beta (TGF-β), the downstream signal transducers (Smads), and transcriptional coactivators p300/CBP in collagen-I synthesis [53]. Upon their receptor-induced activation promoted by TGF-β, Smad2 and Smad3 oligomerize with Smad4 and translocate into the nucleus to regulate the expression of several profibrotic genes. This transcriptional activity is further enhanced by co-activator p300/CBP, which bounds and acetylates Smad2 and Smad3 (Figure 2) [54]. Since the mechanically dependent transcription factor complex YAP/TAZ has been found to cooperate with transcriptional components of the TGF-β pathway [47,55], these data, together, add a novel level of complexity to epigenetic control of cardiac fibrosis through the activity of transcriptional complexes, with histone acetylation activity directly connected to fibrotic paracrine signaling.

## 3. Non-Coding RNAs as Mechanotransducers in Cardiac Fibroblast Differentiation and Fibrosis Progression

Non-coding RNAs (ncRNAs) are RNA species transcribed but not translated into proteins. There are two different categories of ncRNA, depending on size: (1) small non-coding RNAs (<200 bp), including microRNAs (miRNA) and PIWI-interacting RNAs (piRNA); and (2) long non-coding RNAs (lncRNAs, >200 bp), including long intergenic non-coding RNA (lincRNA), circular RNA (circRNAs), natural antisense transcripts (NATs), and enhancer RNAs (eRNAs) [57]. For a complete description of the ncRNAs’ biology in the context of cardiovascular disease, we redirect to a companion article to this report [58]. The principal ncRNAs involved in the cardiac firotic process are listed in Table 1.

To date, several studies demonstrate the involvement of ncRNAs in fibrosis, with potential clinical implications as biomarkers and therapeutic targets. Piccoli and colleagues performed a lncRNA analysis in mice CFs and identified more than 1400 lncRNAs that were deregulated in hypertrophic mice hearts. Among these, CF-specific lncRNA maternally expressed that gene 3 (Meg3) was upregulated [70]. Meg-3 is a conserved chromatin-associated lncRNA that transcriptionally controls the activity of p53 in the promoter regions of MMP-2 and is expressed by CFs. Moreover, it has been reported that MMP-2 is transcriptionally induced by the well-known fibrogenic factor TGF-β. The direct interaction between Meg-3 and p53 activates matrix metalloproteinase-2 (MMP-2), affecting the composition of the ECM. Modification of the cardiac matrix is linked to variations in its mechanical characteristics, which represents further stimuli for the activation of cardiac fibroblasts and fibrosis progression.

An important mechanism involved in the role of miRNAs in cardiac fibrosis has been found from the discovery that several miRNAs are dysregulated in myocardial injury models. One of them, miR-29, was found downregulated in infarcted hearts, with the consequent elevation of profibrotic proteins expression, including collagens [77]. This finding is in line with observations showing a lower expression of the miRNA in fibrosis-related diseases [61], clearly indicating a potential role of miR-29 as a ‘cardioprotector’. On the other hand, compelling evidence shows that expression of miR-29 is under the control of mechanotransduction pathways, and in particular by the YAP/TEAD transcriptional complex through consensus binding sequences for YAP/TEAD in the promoters of the microRNAs-29a/b/c gene variants, thus producing apparently contradictory results. At the moment, it is not possible to conclude whether or not the ischemia-related dowregulation of miR-29 is the only signal promoting cardiac fibrosis in the context of cardiac stiffening. In this regard, it has to be noted that overexpression of miR-29 in fibroblasts has led to only a partial inhibition of collagens expression [77], and that variations in viscoelastic properties of the 3D matrix have stiffness-independent effects on the machinery responsible for miRNA biogenesis [78], thus reconciling the role of YAP as a positive regulator of miR-29 with antifibrotic activity of the miR (Figure 1).

As introduced in the previous section, cells appear to keep a memory of cumulative exposure to mechanical cues, with possible interplay of permanent modifications of the epigenetic landscape [31]. Additionally, ncRNA expression in fibrotic cells may take part in mechanical memory. For example, in mesenchymal cells (MSCs), miR-21 has been identified as a long-term memory keeper of the fibrogenic program [60]. As demonstrated in a study with rats MSCs cultured on gels with different stiffness, the stiffening of the ECM causes the upregulation of miR-21, which is in turn involved in the ECM deposition. The increase in miR-21 expression during stiff-priming depended on the regulation exerted by acutely mechanosensitive myocardin-related transcription factor-A (MRTF-A/MLK-1) and was maintained over two weeks after the removal of mechanical cues. Moreover, silencing miR-21 at the end of stiff-priming resulted in a loss of mechanical memory and the re-acquisition of cellular sensibility into soft matrices [60]. Since MRTF is translocated in the nucleus in response to cell straining [79] and miR-21 is a part of the fibrotic process under the control of TGF-β [56], this evidence establishes a direct link between cell mechanosensing and the ncRNA regulation of cardiac fibrosis (Figure 1).

Recently, a study carried out on mice fibroblasts described a pattern of miRNAs able to exert control on the mechanical properties of cells and tissues [80]. Once they identified the interaction between miRNAs and mRNA, results showed 127 mRNAs. Seventy-three of these specifically encoded a group of proteins called contractility adhesion matrix proteins (CAM). Remarkably, the post-transcriptional regulation of these proteins depended on substrate stiffness and thus on cell mechanics. In order to investigate this regulation, the authors seeded endothelial cells lacking Argonaute2 (AGO2) or DROSHA, two proteins involved in miRNA maturation [72], on substrates with different stiffness. Results showed that, while in control cells, the contractility increased only in high levels of stiffness. In mutant cells, this occurred also on softer substrates. Interestingly, the mutant population exhibited cell area, contractility, and YAP nuclear localization significantly higher compared to controls, thus demonstrating that miRNA biogenesis is necessary for the discrimination between physiologic and non-physiologic cell mechanics.

A further lncRNA emerging for its involvement in fibrosis is the myocardial infarction associated transcript (MIAT) [81], whose expression was reported to be higher in association with cardiac fibrosis in a mouse model of myocardial infarction. As a result of MIAT knockdown, fibrosis was reduced, with a decrease in collagen deposition and inhibition of fibroblasts proliferation. Moreover, cardiac functions improved, with a restoring of physiological levels of fibrosis-related effectors [72].

Finally, there is evidence that the connective tissue growth factor (CTGF), a crucial player in fibrosis onset, is post-transcriptionally controlled by miR-133 and miR-30, which interact directly with the 3′ untranslated region of *CTGF*. In particular, the expression of both miRNAs is inversely related to the levels of CTGF, and their knockdown resulted in an increase of the CTGF amount in mice fibroblasts. Moreover, overexpression of miR-133 or miR-30 reduced CTFG levels and subsequent collagen synthesis [63]. Since CTGF is a canonical target of the YAP/TAZ complex, this constitutes an interesting example of a potential feedback loop for mechanosensing-dependent control of fibrosis in the heart. Almen and collegues unraveled a similar regulatory role of miR-18/19 in CTGF; the expression of these miRs, indeed, resulted in a decrease of collagen type I and III depositions [59].

## 4. Conclusions

In the present report, we highlight the possible interplay of mechanotransduction pathways with epigenetic and ncRNAs in the control of gene expression in heart fibrosis. Cell-based mechanosensing, indeed, has a significant impact on a variety of cellular functions, from embryonic stages to tissue aging and remodeling, and it has a potential to interact on a multilevel and multiscale modality with classical paracrine and metabolic and inflammatory pathways, thus connecting cellular disease memories with tissue architecture and remodeling. While in cancer pathophysiology these concepts are better established and demonstrated, in the cardiovascular diseases scenario, these relationships are only starting to be discovered. We believe that a deeper understanding of these aspects, using interdisciplinary research approaches, will contribute to resolving the molecular setting of cardiac fibrosis and devising therapeutic strategies to combat the increasing burden of heart failure.

## Figures and Tables

**Figure 1 ijms-21-00028-f001:**
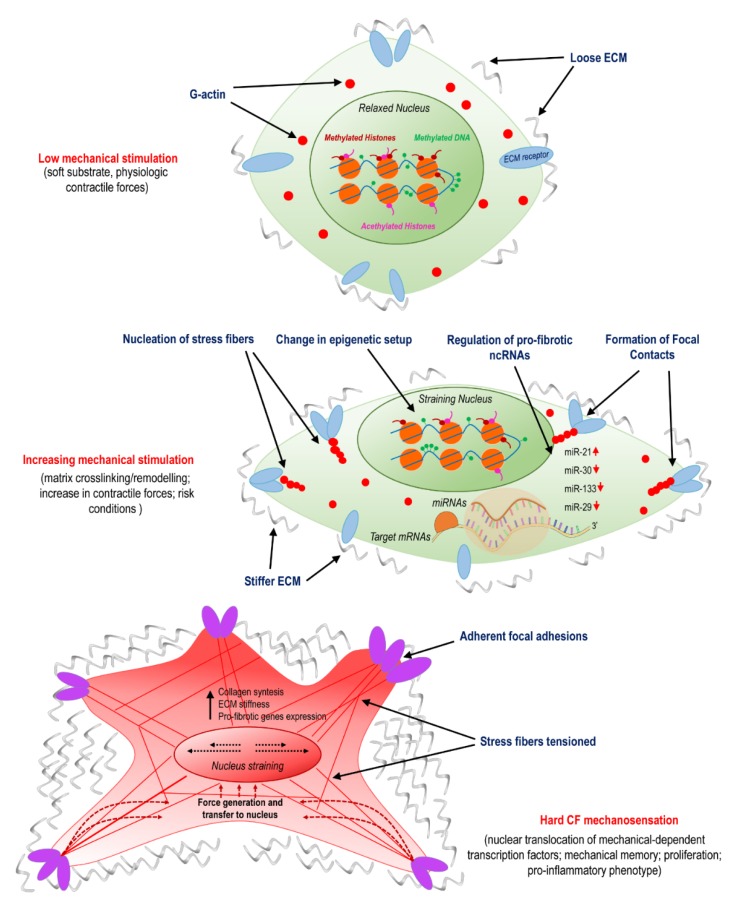
Activation of cardiac fibroblasts (CFs) involves mechanosensing and is related to variations in epigenetic programming and specific non-coding (ncRNAs). The panel illustrates the transition from normal conditions, characterized by a compliant matrix and a low level of cellular mechanical stress; to the pathologic programming occurring due to remodeling/crosslinking of the extracellular matrix (ECM); to the final stage of myo-fibroblast differentiation, which involves nuclear straining, nuclear translocation of mechanically dependent transcription factors, proliferation, and pro-inflammatory phenotypes.

**Figure 2 ijms-21-00028-f002:**
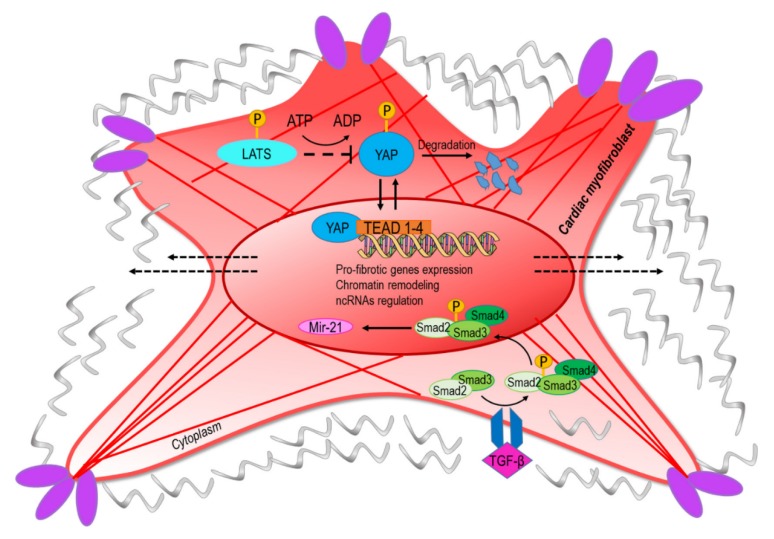
The conversion into myofibroblasts from quiescent fibroblast involves mechanical and paracrine activation of profibrotic pathways. The upper part of the figure represents the transcriptional readout of the YAP/TAZ nuclear translocation, which is dependent on nuclear straining and opening of the nuclear pores [6]. Negative regulation of the pathway is exerted by components of the hippo-kinase pathway (LATS) that phosphorylate YAP and mediate its degradation [55]. The lower part of the panel represents the signaling cascade converging onto miR-21, a classical target of the transforming growth factor-beta (TGF-β)/Smad pathway [56]. The convergence of ‘mechano-paracrine’ pathways affecting the phenotype of profibrotic cells appears to be a new way to integrate signal transduction modalities once considered separated into the control of gene expression and chromatin structure.

**Table 1 ijms-21-00028-t001:** Non-coding (ncRNAs) with implications in cardiac fibrosis.

NcRNA	Target Gene	Pro- or Anti-Fibrotic	References
miR-18	*CTGF*	Anti	[59]
miR-19	*CTGF*	Anti	[59]
miR-21	*PTEN, SMAD7, STAT3, PPAR-* *α*	Pro	[60]
miR-29	*COL1A1, 1A2, 4A5, FBN, ELN1, PDGFR, TAB1, ADAM*	Anti	[61]
miR-130a	*PPAR-α*	Pro	[62]
miR-133	*CTGF*	Anti	[63]
miR-15	*TGF-β, SMAD7, SMAD3*	Anti	[63]
miR-30c	*CTGF*	Anti	[63]
miR-101	*TGF-β*	Anti	[64]
miR-34	*SMAD4*	Pro	[65]
miR-212	*FoxO3*	Pro	[66]
miR-199b	*Dyrk1A*	Pro	[67]
miR-150	*PTX3*	Pro	[68]
miR-155	*SMAD4, SMA2, RhoA*	Pro	[69]
LncRNA-Meg3	*MMP2*	Pro	[70]
LncRNA-Wisper	*Col3α2, Fn1*	Pro	[71]
LncRNA-Miat	*miR-29, miR-30, and miR-133*	Pro	[72]
LncRNA-Malat1	*TGFBR2, SMAD3*	Pro	[73]
LncRNA-PRL	*miR-Let7d*	Pro	[74]
LncRNA-H19	*DUSP5, ERK1*	Pro	[61]
LncRNA-n379519	*miR-30*	Pro	[75]
LncRNA-NR024118	*AT1*	Pro	[76]

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
