# Peer review of "Cell-Based Mechanosensation, Epigenetics, and Non-Coding RNAs in Progression of Cardiac Fibrosis"

_ijms, 2019, doi:10.3390/ijms21010028_

Round 1

Reviewer 1 Report

This manuscript reviews recent findings that link mechanosensation and cardiac fibrosis by going through the involvement of epigenetic and noncoding RNAs. Cell-based mechanosensing in cardiovascular pathophysiology is far to be disclosed, thus this review will shed light on mechanosensing-activated pathways and their role in cardiac diseases.

Author Response

We thank very much the Reviewer for her/his comments. We have carefully proofread the manuscript to make sure that no more spelling mistakes appear in the text.

Reviewer 2 Report

Thank you for the authors for this detailed report on the cellular response to mechanical cues in the cardiac fibrosis context. The authors described the state-of-the art on the main cellular, genetic and epigenetic remodelling taking place in the cardiomyocytes in fibrosis. In general, the manuscript is well written however there are some points that should be addressed to further increase the report quality:

Since the manuscript is submitted under the report category, I think it would be useful for the future readers to have a more detailed figure reflecting the main cellular, genetic and epigenetic remodelling/pathways cited in the manuscript core. For example, the YAP/TAZ complex/pathway since it’s the major one. Figure 1 and table 1 should be referred to in the text Table 1: The label microRNAs (in bold) should not include under it the LncRNAs. I suggest to change the label by ncRNAs so it fits to both. Page 8, the last sentence: “Since modification of the cardiax”, should be corrected to “cardiac”. Page 9, first line “…..this may rapresent”, should be corrected to “represent” Page 9, line 14“…viscoelastic”, should be corrected to “viscoelastic” Page 9, line 24 “maintained”, should be corrected to “maintained”. Page 10, Line 10 “…is necessary for discrimination between physiologic from…”, should be corrected to”…. is necessary for the discrimination between physiologic and…” Page 11, first line “…involved in decryption..”, should be corrected to “…involved in the decryption..” Please try to simplify this sentence.

Author Response

We thank the Reviewer for her/his input that helped us to improve the quality of our manuscript. We addressed all her/his concerns as it follows.

“Since the manuscript is submitted under the report category, I think it would be useful for the future readers to have a more detailed figure reflecting the main cellular, genetic and epigenetic remodelling/pathways cited in the manuscript core. For example, the YAP/TAZ complex/pathway since it’s the major one”

To fulfill Reviewer’s request, we have added a figure where we represented the main mechanism underlying mechanical activation of YAP and its control by the hippo-kinase pathway. The upregulation of miR-21, a typically fibrotic ncRNA is also represented downstream of the TGF-β/Smad pathway.

“Figure 1 and table 1 should be referred to in the text Table 1: The label microRNAs (in bold) should not include under it the LncRNAs. I suggest to change the label by ncRNAs so it fits to both”

The requested changes have been made, thank you

“Page 8, the last sentence: “Since modification of the cardiax”, should be corrected to “cardiac”

The requested changes have been made, thank you

“Page 9, first line “…..this may rapresent”, should be corrected to “represent”

The requested changes have been made, thank you

“Page 9, line 14“…viscoelastic”, should be corrected to “viscoelastic”

The requested changes have been made, thank you

“Page 9 “line 24 “maintained”, should be corrected to “maintained”

The requested changes have been made, thank you

“Page 10, Line 10 “…is necessary for discrimination between physiologic from…”, should be corrected to”…. is necessary for the discrimination between physiologic and…”

The requested changes have been made, thank you

“Page 11, first line “…involved in decryption..”, should be corrected to “…involved in the decryption..” Please try to simplify this sentence”

Sentence has completely reworded and simplified, thank you